# Wearable Core-Shell Piezoelectric Nanofiber Yarns for Body Movement Energy Harvesting

**DOI:** 10.3390/nano9040555

**Published:** 2019-04-04

**Authors:** Sang Hyun Ji, Yong-Soo Cho, Ji Sun Yun

**Affiliations:** 1Energy& Environmental Division, Korea Institute of Ceramic Engineering and Technology, 101, Soho-ro, Jinju 52851, Korea; sanghyun_ji@kicet.re.kr; 2Department of Materials Science & Engineering, Yonsei University, 50, Yonsei-ro, Seodaemun-gu, Seoul 03021, Korea; maname89@naver.com

**Keywords:** lead-free piezoelectric nanofibers, piezoelectric yarns, electrospinning, nanogenerator, wearable devices

## Abstract

In an effort to fabricate a wearable piezoelectric energy harvester based on core-shell piezoelectric yarns with external electrodes, flexible piezoelectric nanofibers of BNT-ST (0.78Bi_0.5_Na_0.5_TiO_3_-0.22SrTiO_3_) and polyvinylidene fluoride-trifluoroethylene (PVDF-TrFE) were initially electrospun. Subsequently, core-shell piezoelectric nanofiber yarns were prepared by twining the yarns around a conductive thread. To create the outer electrode layers, the core-shell piezoelectric nanofiber yarns were braided with conductive thread. Core-shell piezoelectric nanofiber yarns with external electrodes were then directly stitched onto the fabric. In bending tests, the output voltages were investigated according to the total length, effective area, and stitching interval of the piezoelectric yarns. Stitching patterns of the piezoelectric yarns on the fabric were optimized based on these results. The output voltages of the stitched piezoelectric yarns on the fabric were improved with an increase in the pressure, and the output voltage characteristics were investigated according to various body movements of bending and pressing conditions.

## 1. Introduction

In recent years, numerous electronic devices have been developed, including devices that can be worn on the body, such as smart watches, fitness trackers, and even smart fabrics [1,2]. As the demand for these wearable devices has grown, power issues have become a major consideration [3,4,5]. In general, wearable devices with batteries must be lightweight, flexible, stretchable, and highly durable [6,7,8]. Most importantly, the battery should be able to self-generate so that wearable devices can operate without having to be charged externally. Most wearable devices generally use power stored in the form of electrochemical energy in a battery; however, it is not only difficult to make a flexible battery, but also the battery must be replaced or recharged. Some researchers attempted to meet this challenge by proposing a self-charging battery that harvests energy from various sources, such as solar energy, electromagnetic energy, and wind energy [9,10,11]. Batteries charged with such sources are often unstable because they are affected by the surrounding environment as they charge, whereas energy harvesting from body movements have the advantage of being less affected by the surrounding environment. Piezoelectric energy harvesting is proving to be one of the best methods to harvest energy from human body movements. 

There are various piezoelectric ceramic forms, such as bulk, nanowires, and nanofibers [12,13,14]. The bulk form of piezoelectric ceramics has excellent characteristics but no flexibility, restricting its use in wearable devices. On the other hand, nanowires have good flexibility but are difficult to produce in large quantities. In addition, it is difficult to manufacture them using various materials owing to the limited materials that can be manufactured. Piezoelectric nanofibers have advantages when used as an energy harvester in wearable devices due to their simple manufacturing process with good applicability to polymers and piezoelectric ceramics. There have been many studies of piezoelectric energy harvesting using nanofibers [15,16,17]. In these studies, piezoelectric nanofiber modules were created with an encapsulation layer in the polymer films consisting of materials such as polydimethylsiloxane (PDMS) [18], polyamide (PI) [19], or polyethylene terephthalate (PET) [20], though these polymer encapsulation layers generally did not offer a comfortable fit when they were attached to the human body or to clothes. Hence many researchers have fabricated piezoelectric yarns based on piezoelectric nanofibers, and piezoelectric energy harvesters have been developed by directly sewing a wearable device onto fabrics or by fabricating them into woven fabric. 

When making a flexible piezoelectric energy harvester, how the two electrode layers are applied to the piezoelectric nanofiber is critical because if the electrode layers are not flexible and do not become fibrous, there will be an uncomfortable fit when the harvester is attached to the human body or clothes. Durability issues may also arise. To address these issues, we designed a wearable piezoelectric energy harvester based on core-shell piezoelectric yarns with external electrodes, as shown in Figure 1. Electrospun piezoelectric nanofibers in the shell parts were twined around a conductive thread (the inner electrode layer) in the core parts, and the core-shell piezoelectric nanofiber yarns were then braided with two strands of conductive thread for the outer electrode layers. The outer electrode layers were braided with the core-shell piezoelectric nanofiber yarns, not the coating, which improved the performance and durability of the energy harvesting process owing to the shapes of the module, which are deformed according to various body movements. Moreover, spaces due to the separation of the outer electrode layers are mostly not generated, and the outer electrode layers are integrally deformed with the core-shell piezoelectric nanofiber yarns. The core-shell piezoelectric yarns with external electrodes were directly stitched onto fabric to create a wearable piezoelectric nanofiber energy harvester module, and the output voltage performance capabilities were investigated according to various bending and pressing body movements.

## 2. Experimental Procedure

The precursor solution for the electrospinning of the BNT-ST (0.78Bi_0.5_Na_0.5_TiO_3_-0.22SrTiO_3_) nanofibers used in this study was prepared from lead-free BNT-ST piezoelectric ceramic powders created via a solid-state reaction method. The raw materials were high purity Bi_2_O_3_, Na_2_CO_3_, SrCO_3_, and TiO_3_ powders (Kojundo Chemical, 99.9%, Saitama, Japan), and the particle size of the BNT-ST ceramics was adjusted to less than 100 μm. Polyvinylidene fluoride-trifluoroethylene (PVDF-TrFE) copolymer (VDF 75%, TrFE 25%; Measurement Specialties, Hamptom, VA, USA) was vigorously stirred in *N*-*N*’-dimethylformamide (DMF, 99.5%; Sigma-Aldrich, St. Louis, MO, USA) and acetone (99.9%; Sigma-Aldrich) for 24 h at room temperature at a 2:2:5 weight ratio (PVDF-TrFE to acetone to DMF). After stirring, 60 wt% of BNT-ST powder was added to the PVDF-TrFE precursor solution with subsequent vigorous stirring at room temperature for 24 h to obtain a homogeneous solution [21].

Ten milliliters of the BNT-ST/PVDF-TrFE precursor solution was placed in a syringe with a 21 G metallic needle, and the needle was connected to a voltage power supplier. The flow rates for electrospinning were controlled using syringe pumps at 1 mL/h, and the distance between the tip and the collector was fixed at 10 cm. The applied voltage was controlled at 10–15 kV. The nanofibers were collected on a drum collector rotating at a speed of 1500 rpm to align the nanofibers [14].

The electrospun BNT-ST/PVDF-TrFE nanofibers were manually twined around a conductive thread with a diameter of about 0.1 mm, and the total diameter of the core-shell piezoelectric nanofiber yarns was approximately 0.3 mm. An annealing process conducted at a temperature of 70 °C and lasting nearly one minute was used to stabilize the yarns. The outer electrode layers of the two conductive thread strands were formed by braiding them, as shown in Figure 1. The microstructure of the core-shell piezoelectric yarns with the external electrodes was analyzed using an optical microscope (Olympus BX51 microscope, Tokyo, Japan).

To investigate the output voltages during the bending tests, the core-shell piezoelectric yarns with the external electrodes were directly stitched onto fabric (7 cm × 5 cm) with a different total length, effective area, and stitching interval of the piezoelectric yarns. Afterwards, high voltages of 1 kV at room temperature for 1 h as a polling process were applied to the wearable piezoelectric nanofiber energy harvester module [22]. The output voltages according to bending movements of 1 Hz by a bending machine (SPG Co., Ltd, Incheon, Korea) were recorded with an oscilloscope (Wavejet322, LeCroy, New York, NY, USA). 

With regard to the output voltages according to the applied pressure, a custom-made striking machine with a constant striking force to the effective area was used. The striking force of the machine was changed according to the operating angle, and the output force was calculated as follows:(1)$$N\;\left(kg\cdot m/s^{2}\right)=6.3x+68$$
Here, *x* is the operating angle of the striking machine. Core-shell piezoelectric yarns with external electrodes were directly stitched onto fabric with an area of 0.7 cm^2^ with intervals of 0.15 cm, and the output voltage was measured according to the pressure (operating angle) using an oscilloscope.

The output voltages (*V*) of core-shell piezoelectric yarns with external electrodes directly stitched onto fabric based on optimized stitching patterns were observed using an oscilloscope according to the various bending and pressing types of body movements. The output voltages (*V*) of a wearable device module connected to resistance (*R*) of 5 MΩ were measured, and output powers (*P*) and output currents (*I*) were calculated based on the basic formula of *P* = *V*^2^/*R* = *VI* [23,24].

## 3. Results and Discussion

In an effort to optimize the stitching patterns of core-shell piezoelectric yarns with external electrodes during bending tests, the output voltages were observed according to the total length, effective area, and stitching interval of the core-shell piezoelectric yarns, as shown in Figure 2. The inset images of Figure 2a depict optical images of stitched core-shell piezoelectric yarns with lengths of 5, 10, and 15 cm. The output voltages of the core-shell piezoelectric yarns with lengths of 5, 10, and 15 cm were gradually increased to 2.2, 4.7, and 5.4 mV at 1 Hz of bending motion, as shown in Figure 2a. As expected, as the length of the piezoelectric yarns was increased, the amount of piezoelectric nanofibers increased, and more energy could be harvested. To achieve this result, it was necessary to stitch the piezoelectric yarns as long as possible in a fixed area to improve the amount of energy which could be harvested. As shown in the first and third optical images in Figure 2b, for piezoelectric yarns with the same total length of 15 cm stitched with different module widths of 3 and 5 cm, the output voltage (16.4 mV) for the module with the 3 cm width was higher than the output voltage (8.4 mV) of the module with the 5 cm width. As shown in the first and second optical images in Figure 2b, for the same length (9 cm) with the effective width of 3 cm, although the piezoelectric yarns had different total lengths of 15 and 9 cm, these piezoelectric modules had similar output voltages of 8.4 and 8.8 mV, respectively. In other words, longer piezoelectric yarns in the effective area stimulated by bending motion allowed more energy to be harvested. The piezoelectric yarns outside the effective area did not play a role in energy harvesting. Figure 2c shows that in core-shell piezoelectric yarns with the same total length (15 cm) and the same module width (3 cm), the output voltages gradually increased to 8.8, 16.4, and 19.1 mV as the stitching intervals were correspondingly decreased from 0.5 to 0.3 cm and finally to 0.15 cm. The surface of the piezoelectric yarn was not completely covered by the external electrodes due to the characteristics of the braiding process, and an area was exposed due to the external electrodes. We expected that as the stitching interval was decreased, the exposed surface of the piezoelectric yarn was also influenced by the external electrodes of the subsequent yarns, and more energy was harvested. To improve the level of energy harvesting during the bending tests, the stitching patterns of the core-shell piezoelectric yarns were optimized such that the piezoelectric yarns could be stitched with lengths as long as possible with a stitching interval of 0.15 cm in the effective area stimulated during the bending motion. 

Subsequently, we investigated the output voltages according to the applied pressure. Figure 3a shows the custom-made striking machine used to apply constant pressure. Core-shell piezoelectric yarns with external electrodes were directly stitched at intervals at 0.15 cm onto fabric to match the struck area of 0.7 cm^2^. The force was calculated according to the angle based on Equation (1), and the pressure was calculated by the force applied to the struck area of 0.7 cm^2^. The pressure increased by about 4.4 kPa when the angle was increased by 5°. The output voltages were observed according to the applied pressure, as shown in Figure 3b; as the pressure was increased, the output voltage increased with a slope of 0.0047, showing a maximum output voltage of 0.181 V at a pressure of 47.77 kPa. However, the output voltage dropped sharply to 0.004 V at a pressure of 51.27 kPa. Under these conditions, we expect that these wearable energy harvester modules based on the core-shell piezoelectric yarns with external electrodes will be usable for energy harvesting at pressures less than 50 kPa. 

In an investigation of the energy harvesting performance based on body bending movements, the wearable energy harvester modules based on core-shell piezoelectric yarns with external electrodes were stitched with a stitching interval of 0.15 cm in the effective area stimulated under bending motion into various garments—in this case, a glove and elbow and knee guards, as shown in Figure 4. In-bending movements refers to bending movements conducted when the module is located on the inside area of the bend, and out-bending movements refers to bending movements conducted when the module is located on the outside area of the bend. The peak-to-peak output voltages (currents) were 1.6 V (0.32 μA) and 2.2 V (0.44 μA) according to finger movements in the in-bending and out-bending parts, respectively, when a cotton glove was worn that was stitched in the finger parts with the core-shell piezoelectric yarns with a length of 205 cm, as shown in Figure 4a. After the guard stitched in the bending parts with the core-shell piezoelectric yarns with a length of 200 cm was worn, the peak-to-peak output voltages (1.7 V) and peak-to-peak output currents (0.34 μA) according to elbow movements of the in-bending part were higher than the peak-to-peak output voltages (1.1 V) and currents (0.22 μA) for the out-bending movement, as shown in Figure 4b. The output voltages according to knee movements were similar to those of elbow movements, as shown in Figure 4c. The output voltages and output currents according to the in-bending movements were generally higher than those of the out-bending movements, as the in-bending motions involved more inward folding. It was also observed that a greater amount of energy was harvested during small bending movements, such as movements by fingers than during large bending movements such as those by elbows and knees. Furthermore, our results confirmed that maximum peak-to-peak power of about 0.97 μW can be harvested from wearable piezoelectric yarns by body bending movements, for instance finger, elbow, and knee movements.

To investigate the energy harvesting performance based on body pressure movement, the wearable energy harvester modules based on the core-shell piezoelectric yarns with external electrodes were directly stitched at a stitching interval of 0.15 cm in an effective area stimulated by pressure motions into various garments—here, a glove and a shoe insole, as shown in Figure 5. The effective area was determined by referring to pressure patterns generated according to clapping movements [25], and core-shell piezoelectric yarns with a length of 289 cm were stitched into the effective area of a cotton glove. After the cotton glove was worn, peak-to-peak output voltage of 2.1 V and peak-to-peak output current of 0.42 μA were measured according to clapping movements, as shown in Figure 5a. Furthermore, in the effective area of a shoe insole determined by the pressure patterns under the right foot [26], core-shell piezoelectric yarns with a length of 256 cm were directly stitched, and peak-to-peak output voltages (currents) of 0.7 V (0.14 μA) and 1.9 V (0.38 μA) were observed according to walking and running movements, respectively, by a person with a weight of 82 kg, as shown in Figure 5b. Higher output voltages were generated from running as compared to walking because more pressure is applied during running. This experiment confirmed that maximum peak-to-peak power of approximately 0.88 μW can be harvested from wearable piezoelectric yarns by body pressure movements, particularly when clapping and running.

## 4. Conclusions

Flexible piezoelectric nanofibers were prepared by an electrospinning process of BNT-ST and PVDF-TrFE composite solutions, and piezoelectric nanofibers were twined around a conductive thread to create the core-shell piezoelectric nanofiber yarns. Outer electrode layers were applied by braiding the core-shell piezoelectric nanofiber yarns with conductive thread, and a wearable piezoelectric energy harvester based on core-shell piezoelectric yarns with external electrodes was finally fabricated by directly stitching it onto fabric. The output voltages during bending tests were investigated according to the total length, effective area, and stitching interval of the piezoelectric yarns, and optimized stitching patterns of the core-shell piezoelectric yarns were made as long as possible with a stitching interval of 0.15 cm inside an effective area. The output voltages were investigated according to the applied pressure, and as the pressure was increased, the output voltage increased, with a maximum available pressure of approximately 50 kPa. As various garments (a glove, a guard, and a shoe insole) directly stitched with the core-shell piezoelectric yarns were worn, energy could be harvested according to various bending and pressing body movements.

## Figures and Tables

**Figure 1 nanomaterials-09-00555-f001:**
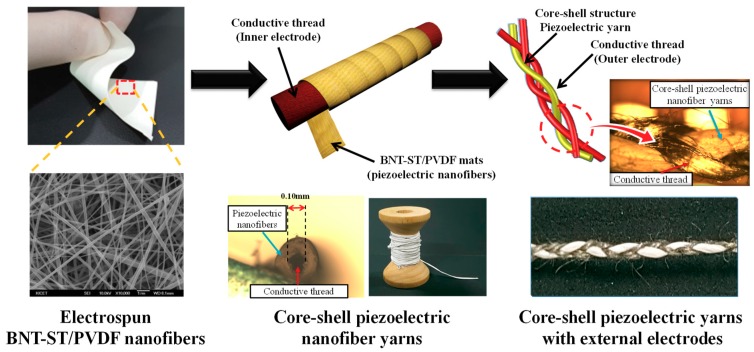
Schematic diagrams of the fabrication process of the core-shell piezoelectric yarns with external electrodes.

**Figure 2 nanomaterials-09-00555-f002:**
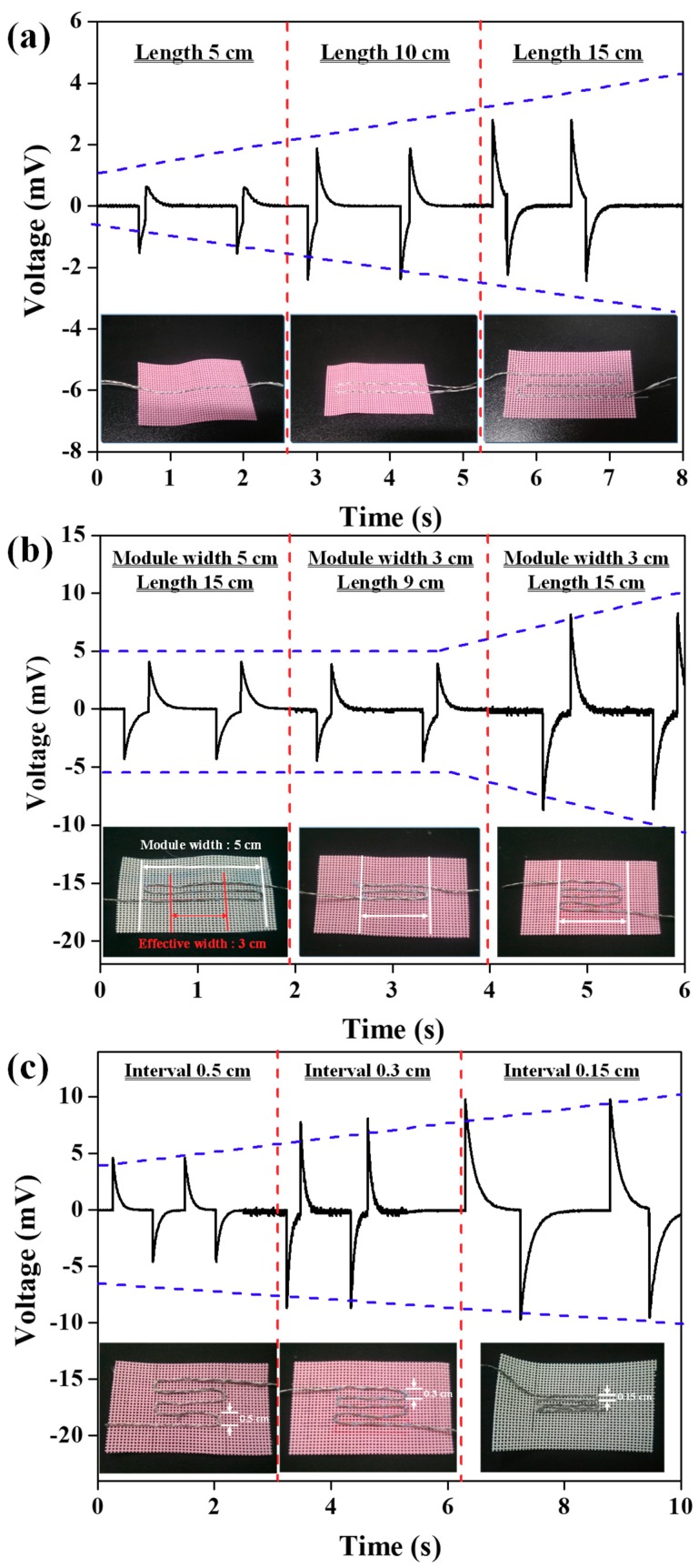
Output voltages according to the (**a**) total length, (**b**) effective area, (**c**) and stitching interval of core-shell piezoelectric yarns during bending tests.

**Figure 3 nanomaterials-09-00555-f003:**
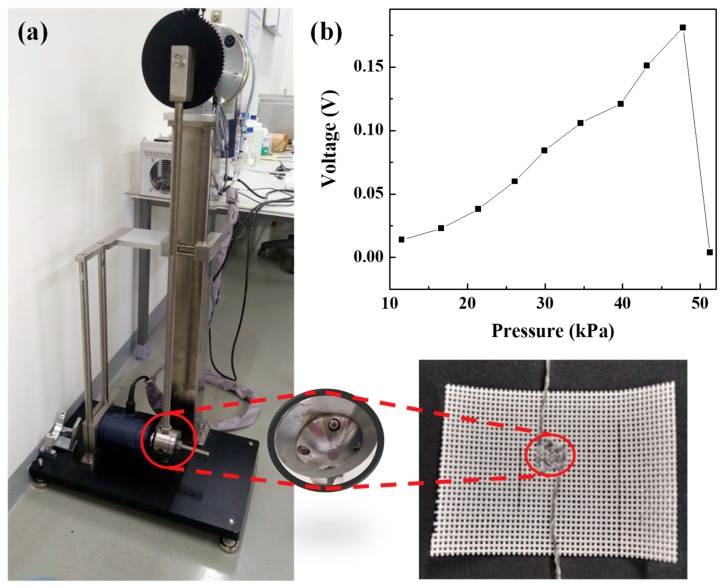
(**a**) Equipment used to apply constant pressure to a piezoelectric energy harvester module stitched with a width of 0.7 cm^2^ and interval of 0.15 cm by the core-shell piezoelectric yarns, and (**b**) output voltage according to the applied pressure.

**Figure 4 nanomaterials-09-00555-f004:**
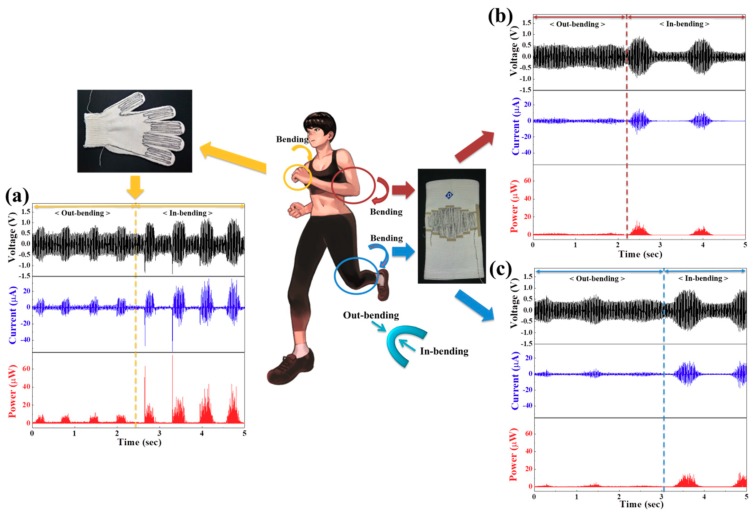
(**a**) Output voltages, output currents, and output powers according to in-bending and out-bending finger movements as a cotton glove stitched in the finger parts with the core-shell piezoelectric yarns is worn, and output voltages, output currents, and output powers according to in-bending and out-bending (**b**) elbow and (**c**) knee movements as the guard stitched in the elbow and knee parts by the core-shell piezoelectric yarns is worn, respectively.

**Figure 5 nanomaterials-09-00555-f005:**
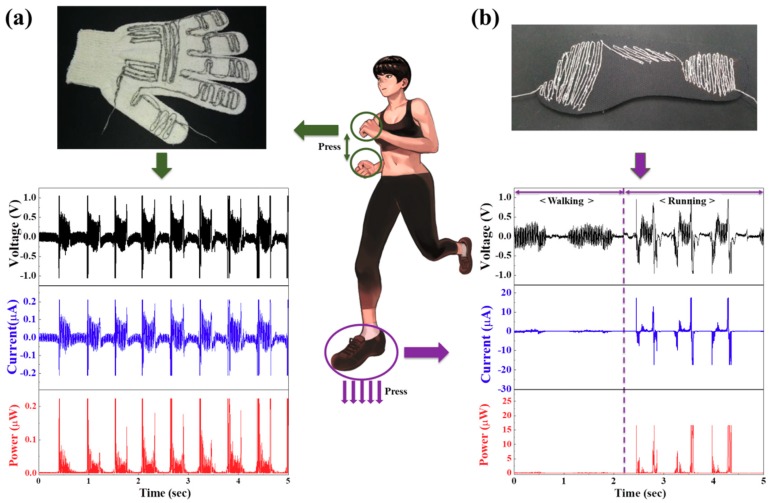
(**a**) Output voltages, output currents, and output powers according to clapping movements as a cotton glove stitched in the effective area with the core-shell piezoelectric yarns is worn, and (**b**) output voltages, output currents, and output powers according to walking and running movements as a shoe insole stitched in the effective area with the core-shell piezoelectric yarns is worn.

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
