# Peer review of "Wearable Core-Shell Piezoelectric Nanofiber Yarns for Body Movement Energy Harvesting"

_nanomaterials, 2019, doi:10.3390/nano9040555_

Round 1

Reviewer 1 Report

The research presented in the manuscript addresses the issue of harvesting energy from body movements  using wearable core-shell piezoelectric nanofiber yarns.

Two basic things should be improved:

1. The core-shell piezoelectric nanofiber yarns structure should be well described , and a schematic drawing should be added to understand the location of the inner and outer electrodes. This issue is crucial for the understanding of the proposed method.

2. Trying to measure the voltage is misleading from the output point of view. It is known that while the voltage generated by a piezoelectric device might be high, its current is very low. Therefore the output results should present energy or power and not voltage. As such the authors are requested to revise their graphs and present their results in a conformal way.

3. Another issue is how the voltage and then the power was measured ? Was it measure on an outside connected resistor or capacitor ? Please provide details on this issue.

Author Response

1. The core-shell piezoelectric nanofiber yarns structure should be well described, and a schematic drawing should be added to understand the location of the inner and outer electrodes. This issue is crucial for the understanding of the proposed method.

è Thank you for your kind comments. According to your comments, I revised the schematic drawing in Fig. 1 as follows:

Fig. 1. Schematic diagrams of the fabrication process of the core-shell piezoelectric yarns with external electrodes

2. Trying to measure the voltage is misleading from the output point of view. It is known that while the voltage generated by a piezoelectric device might be high, its current is very low. Therefore the output results should present energy or power and not voltage. As such the authors are requested to revise their graphs and present their results in a conformal way.

è Thank you for your kind comments. According to your comment, I have added a graph of power results in Fig.4 and Fig.5 as follows:.

Fig. 4. (a) Output voltages, output currents and output powers according to in-bending and out-bending finger movements as a cotton glove stitched in the finger parts with the core-shell piezoelectric yarns is worn, and output voltages, output currents and output powers according to in-bending and out-bending (b) elbow and (c) knee movements as the guard stitched in the elbow and knee parts by the core-shell piezoelectric yarns is worn, respectively.

Fig. 5. (a) Output voltages, output currents and output powers according to clapping movements as a cotton glove stitched in the effective area with the core-shell piezoelectric yarns is worn, and (b) output voltages, output currents and output powers according to walking and running movements as a shoe insole stitched in the effective area with the core-shell piezoelectric yarns is worn.

3. Another issue is how the voltage and then the power was measured ? Was it measure on an outside connected resistor or capacitor?  Please provide details on this issue.

è. Thank you for your kind comments. According to your comment, I further explained as follows:
“The output voltages (V) of core-shell piezoelectric yarns with external electrodes directly stitched onto fabric based on optimized stitching patterns were observed using an oscilloscope according to the various bending and pressing types of body movements. The output voltages (V) of a wearable device module connected to resistance (R) of 5 MΩ were measured, and output powers (P) and output currents (I) were calculated based on the basic formula of P = V2/R = VI [23,24].”

Reviewer 2 Report

The paper presents a wearable core-shell piezoelectric nanofiber for body movement energy harvesting. The paper is well written and clear except on one point: output powers.

You mention that output powers are calculated with R=5MOhms. However, output voltages, currents and powers mentioned in the paper do not match with P=U*I=U^2/R=R*I^2.

Please check that mean output powers are mentioned and not instantaneous output power Pinstantaneous=Umax(open circuit)*Imax(short circuit), which is a nonsense in energy harvesting devices.

Output powers, currents and voltages must be checked throughout the paper before it can be published.

Author Response

The paper presents a wearable core-shell piezoelectric nanofiber for body movement energy harvesting. The paper is well written and clear except on one point: output powers.

You mention that output powers are calculated with R=5MOhms. However, output voltages, currents and powers mentioned in the paper do not match with P=U*I=U^2/R=R*I^2.

Please check that mean output powers are mentioned and not instantaneous output power Pinstantaneous=Umax(open circuit)*Imax(short circuit), which is a nonsense in energy harvesting devices.

Output powers, currents and voltages must be checked throughout the paper before it can be published.

è Thank you for your kind comments. According to your comment, I have recalculated the output power and corrected it. I used the most common methods to measure power values in a piezoelectric energy harvester [23,24]. I checked the output power values carefully, and I obviously explained as follows:

“The output voltages (V) of core-shell piezoelectric yarns with external electrodes directly stitched onto fabric based on optimized stitching patterns were observed using an oscilloscope according to the various bending and pressing types of body movements. The output voltages (V) of a wearable device module connected to resistance (R) of 5 MΩ were measured, and output powers (P) and output currents (I) were calculated based on the basic formula of P = V2/R = VI [23,24].”

Round 2

Reviewer 1 Report

The authors provided the missing information, regarding how was the voltage measured. They quote a resistor of  5MΩ. 

If this is the number than, the results written in lines 169,170 are wrong:

"The output voltages (currents) were 2.0 V (55.1 μA) and 1.4  V (27.3 μA) according to finger movements"  !!!

I=V/R=2/5MΩ=0.4μA; 1.4/5MΩ=0.28μA

What happened ? Look also in Figs. 4 and 5 and comment on it. 

Author Response

è Thank you for your kind comments. It is highly regrettable that mistakes are repeated. According to your comments, I have recalculated the output voltages and currents, and corrected it in results and discussion part, and in Fig 4 and Fig 5. Furthermore, I obviously explained as follows:

 “The peak-to-peak output voltages (currents) were 1.6 V (0.32
μA) and 2.2 V (0.44 μA) according to finger movements in the in-bending and out-bending parts, respectively, when a cotton glove was worn that was stitched in the finger parts with the core-shell piezoelectric yarns with a length of 205 cm, as shown in Fig. 4 (a). After the guard stitched in the bending parts with the core-shell piezoelectric yarns with a length of 200 cm was worn, the peak-to-peak output voltages (1.1 V) and peak-to-peak output currents (0.22 μA) according to elbow movements of the in-bending part were higher than the peak-to-peak output voltages (1.7 V) and currents (0.34 μA) for the out-bending movement, as shown in Fig. 4 (b).”

“Furthermore, our results confirmed that maximum peak-to-peak power of about 0.97 μW can be harvested from wearable piezoelectric yarns by body bending movements, for instance finger, elbow and knee movements.”

 “and core-shell piezoelectric yarns with a length of 289 cm were stitched into the effective area of a cotton glove. After the cotton glove was worn, peak-to-peak output voltage
of 2.1 V and peak-to-peak output current of 0.42 μA were measured according to clapping movements, as shown in Fig. 5 (a).”

“core-shell piezoelectric yarns with a length of 256 cm were directly stitched, and peak-to-peak output voltages (currents) of 0.7 V (0.14 μA) and 1.9 V (0.38 μA) were observed according to walking and running movements, respectively, by a person with a weight of 82 kg, as shown in Fig. 5 (b).”

“This experiment confirmed that maximum peak-to-peak power of approximately 0.88 μW can be harvested from wearable piezoelectric yarns by body pressure movements, particularly clapping and running.”

Reviewer 2 Report

The way the output power is computed is still unclear. 

You mention that 110µW can be harvested in the paper. Yet, none of your figures (Fig 4, Fig 5) show output powers reaching more than 40µW. Moreover, 40µW is a peak power, not a mean output power.

Please provide mean output powers (and peak output powers if you want). But, you have to mention clearly the output power you present: is it a peak output power or a mean output power.

I still recommend major revisions until this issue is not solved.

Author Response

è Thank you for your kind comments. It is highly regrettable that mistakes are repeated. According to your comments, I have recalculated the output voltages and currents, and corrected it in results and discussion part, and in Fig 4 and Fig 5. Because in the piezoelectric energy harvesting fields, the peak-to-peak voltage, current and power are generally used, I wrote these values in results and discussion part. I obviously explained as follows:

 “The peak-to-peak output voltages (currents) were 1.6 V (0.32 μA) and 2.2 V (0.44 μA) according to finger movements in the in-bending and out-bending parts, respectively, when a cotton glove was worn that was stitched in the finger parts with the core-shell piezoelectric yarns with a length of 205 cm, as shown in Fig. 4 (a). After the guard stitched in the bending parts with the core-shell piezoelectric yarns with a length of 200 cm was worn, the peak-to-peak output voltages (1.1 V) and peak-to-peak output currents (0.22 μA) according to elbow movements of the in-bending part were higher than the peak-to-peak output voltages (1.7 V) and currents (0.34 μA) for the out-bending movement, as shown in Fig. 4 (b).”

“Furthermore, our results confirmed that maximum peak-to-peak power of about 0.97 μW can be harvested from wearable piezoelectric yarns by body bending movements, for instance finger, elbow and knee movements.”

 “and core-shell piezoelectric yarns with a length of 289 cm were stitched into the effective area of a cotton glove. After the cotton glove was worn, peak-to-peak output voltage
of 2.1 V and peak-to-peak output current of 0.42 μA were measured according to clapping movements, as shown in Fig. 5 (a).”

“core-shell piezoelectric yarns with a length of 256 cm were directly stitched, and peak-to-peak output voltages (currents) of 0.7 V (0.14 μA) and 1.9 V (0.38 μA) were observed according to walking and running movements, respectively, by a person with a weight of 82 kg, as shown in Fig. 5 (b).”

“This experiment confirmed that maximum peak-to-peak power of approximately 0.88 μW can be harvested from wearable piezoelectric yarns by body pressure movements, particularly clapping and running.”
